# Healthcare Workers’ Perspectives of mHealth Adoption Factors in the Developing World: Scoping Review

**DOI:** 10.3390/ijerph20021244

**Published:** 2023-01-10

**Authors:** Michael Addotey-Delove, Richard E. Scott, Maurice Mars

**Affiliations:** 1Department of TeleHealth, School of Nursing & Public Health, College of Health Sciences, University of KwaZulu-Natal, Durban 4041, South Africa; 2Department of Community Health Sciences, Cumming School of Medicine, University of Calgary, Calgary, AB T2N 4Z6, Canada; 3College of Nursing and Health Sciences, Flinders University, Adelaide, SA 5042, Australia

**Keywords:** mHealth, adoption, mhealth adoption, framework, healthcare workers, developing countries, telemedicine

## Abstract

Background: mHealth applications provide health practitioners with platforms that enable disease management, facilitate drug adherence, facilitate drug adherence, speed up diagnosis, monitor outbreaks, take and transfer medical images, and provide advice. Many developing economies are investing more in mobile telecommunication infrastructure than in road transport and electric power generation. Despite this, mHealth has not seen widespread adoption by healthcare workers in the developing world. This study reports a scoping review of factors that impact the adoption of mHealth by healthcare workers in the developing world, and based on these findings, a framework is developed for enhancing mHealth adoption by healthcare workers in the developing world. Methods: A structured literature search was performed using PubMed and Scopus, supplemented by hand searching. The searches were restricted to articles in English during the period January 2009 to December 2019 and relevant to the developing world that addressed: mobile phone use by healthcare workers and identified factors impacting the adoption of mHealth implementations. All authors reviewed selected papers, with final inclusion by consensus. Data abstraction was performed by all authors. The results were used to develop the conceptual framework using inductive iterative content analysis. Results and Discussion: Of 919 articles, 181 met the inclusion criteria and, following a review of full papers, 85 reported factors that impact (promote or impede) healthcare worker adoption of mHealth applications. These factors were categorised into 18 themes and, after continued iterative review and discussion were reduced to 7 primary categories (engagement/funding, infrastructure, training/technical support, healthcare workers’ mobile—cost/ownership, system utility, motivation/staffing, patients’ mobile—cost/ownership), with 17 sub-categories. These were used to design the proposed framework. Conclusions: Successful adoption of mHealth by healthcare workers in the developing world will depend on addressing the factors identified in the proposed framework. They must be assessed in each specific setting prior to mHealth implementation. Application of the proposed framework will help shape future policy and practice of mHealth implementation in the developing world and increase adoption by health workers.

## 1. Introduction

Application of mHealth (“the use of mobile wireless technologies for health” [1]) has revolutionised healthcare to a point that there are few facets of medical practice in which mHealth systems have not been implemented or considered [2]. mHealth can enhance service delivery [3], disease management [4], patient monitoring [5], mental health care [6], post-operative care [7], the sharing of images and reports using common apps [8], promote drug treatment adherence and behaviour change [9], and facilitate patient communication [10]. Data collection using mobile devices has been shown to be more efficient and reliable than traditional pen-and-paper methods [11], and its benefits in supply management have been explored [12]. Increasingly the benefits of mobile learning (mLearning) to support healthcare worker training are being explored [13,14]. As seen in the COVID-19 pandemic, mHealth has facilitated remote patient monitoring [15], coordination and monitoring of vaccination campaigns [16], communication, contact tracing, and training [17].

The benefits of mHealth are particularly relevant in the developing world [1] with its shortages of health professionals and disproportionate burden of disease, but its uptake has been slow. Potential adoption of mHealth has typically been assessed using predetermined theories with existing models or frameworks, most frequently the Technology Acceptance Model (TAM), TAM2, the Unified Theory of Acceptance and Use of Technology (UTAUT), and UTAUT 2. Such prior studies have resulted in a large number of investigated variables and a growing number of adapted technology acceptance theories [18]. These approaches tend to be formulaic and have been noted to have limitations. They do not identify the full spectrum of factors that impact technology adoption in the health sector [19], and therefore do not account for 100% of variance in behavioural intention to adopt a technology, but anywhere between 12 and 95%, on average ~54% [20,21] (Table 1).

There is growing concern regarding application of the models. For example, they have been suggested to take a narrow perspective and to be overly simplistic [22]. Of particular note is that TAM and UTAUT have failed to “provide stable predictive capabilities for acceptance and use of technologies in health care”, presumed to be associated with the complexity of the healthcare setting [23]. Further, most studies apply an extension of the original model, which has led to the suggestion that no optimal version of TAM exists for application to health services [24].

Importantly, it has been noted that these models do not accept the subjective norm and only fit circumstances where use is voluntary, and users are experienced [25]; in the healthcare setting technology use is typically mandatory, and users are often ‘first timers’. In addition, issues seldom noted in studies have arisen; for example, healthcare workers have noted concern about the time some patients contact them for services and similarly, patients’ desire to receive health-related messages at convenient times, with specific frequency, or preference for different delivery channels [26], and in their own language [27]. They also do not typically take into account equity and equality issues such as gender and education levels influencing phone ownership and Internet access [28]. As a consequence of these limitations, an alternate approach to determining factors that impact mHealth adoption and use is needed.

**Table 1 ijerph-20-01244-t001:** Reported explanation of variance in the intention to use a technology solution in the health sector (2000 to 2021, inclusive).

Range (%)	Reported % Variance
81–100	81% [29], 95% [30], 68% [31]
61–80	62% [32], 65% [33], 69% [34]
41–60	51% [35], 53% [36], 51% [37], 41% [38], 59% [39], 59% [40], 60% [41], 52% [42], 56% [42], 48% [43]
21–40	28% [44], 28% [45], 38% [46], 32% [47]
Other reported range	34–52% [48], 20 or 44% [49], 12 or 46% [50]

eHealth is considered crucial in facilitating Universal Health Coverage (UHC) [51,52], and mHealth has been considered to be essential for facilitating access to healthcare for underserved populations, particularly in rural and remote locations [53,54]. Given the demonstrated breadth of mHealth applications, there is need to investigate and develop approaches that will facilitate and increase its widespread adoption, especially in those countries most in need of improved healthcare provision.

The COVID-19 pandemic has necessitated the rapid adoption of eHealth (including mHealth), raising general awareness [55,56], but in most instances, without the formal approaches usually considered necessary for eHealth implementation and adoption [57]. This has been facilitated by relaxation of laws and regulations, and the trend is anticipated to continue post-pandemic [58,59]. A framework is required to identify and categorise those factors that have been shown, through review of actual implementations in the developing world, to impact adoption by healthcare workers of mHealth. A scoping review would provide the breadth of understanding necessary.

The aims of this study were to identify and examine empirical evidence to answer the research question “what factors have impacted (enabled or impeded) adoption of mHealth by healthcare workers in developing countries” and then use the findings to develop an evidence-based and pragmatic framework. The framework and study findings will contribute to further understanding of technology adoption, and will inform policy and help facilitate the future successful adoption of mHealth in the developing world.

## 2. Methods

A scoping review was performed in January 2020 according to the five steps of Arksey and O’Malley [60] and in accordance with PRISMA-SR guidelines. ‘PRISMA-SR’ refers specifically to the PRISMA Scoping Review guidelines. ‘PRISMA’ refers to systematic reviews and meta-analyses. The search was limited to January 2020 to avoid the influence of the forced implementation caused by the COVID-19 pandemic. Two databases were chosen: PubMed (life sciences and biomedical resources) and Scopus (health science, life science, and social science resources). Each was searched for papers on factors facilitating or impeding mHealth adoption by healthcare workers in the developing world, published to 31 December 2019. The following search string was used for PubMed: (“telemedicine”[MeSH Terms] OR “telemedicine”[All Fields] OR “mHealth”[All Fields] OR “cell phones”[MeSH Terms]) AND (“developing countries” [MeSH Terms] OR “Africa”[MeSH Terms] OR “Asia”[MeSH Terms] OR “Latin America”[MeSH Terms]) AND ((“Health personnel” [MeSH Terms]) OR (“barriers”[All Fields]) OR (“barrier”[All Fields]) OR (“challenges”[All Fields]) OR (“facilitators”[All Fields]) OR (“successes”[All Fields]) OR (“obstacles”[All Fields]) OR (“obstacle”[All Fields]) OR (“failure”[All Fields]) OR (“success”[All Fields])). Similarly, a comprehensive search string was used for Scopus. Handsearching was performed through snowballing using reference listings from included studies.

The inclusion criteria were—the resource was original research in English, published after 2008, and addressed: mobile phone use by healthcare workers or healthcare professionals; identified factors impacting adoption of mHealth implementations; and took place in the developing world. The exclusion criteria were—the resource addressed telemedicine or telehealth more broadly; took place in the developed world; or solely addressed factors impacting mHealth adoption by stakeholders other than healthcare workers. The searches were consolidated within EndNote and duplicates removed. Titles and abstracts of remaining articles were reviewed against inclusion and exclusion criteria and studies selected for full text review by two authors (MAD, RES), with differences of opinion settled by consensus with the third author (MM). The full-text articles were then reviewed, summarised, and data charted in an Excel spreadsheet (author(s), title, date of publication, country, mHealth type, study method, factors impacting adoption).

Inductive iterative content analysis was used to independently categorise elements, and extract themes from the data. A published approach to content analysis was adopted and no underlying framework for categorisation was used. Two researchers (MAD, RES) independently read and [61] re-read the selected literature resources (study data) to identify factors that the researchers believed were important and might impact a healthcare workers adoption of m-health. This process gave rise to initial groupings. These factors were collated in an Excel database, then iteratively and collaboratively reviewed and agreement reached in further grouping them into distinct categories. These categories dealt with the same or related issue(s) and were assigned a descriptive title. Thereafter, categories were again iteratively and collaboratively appraised, and agreement reached on placing multiple categories into groupings to identify common themes, and each assigned a descriptive title. This process gave rise to a final set of themes and subcategories [61].

## 3. Results

PubMed identified 702 resources, whilst Scopus identified 199 resources, and 18 were found through handsearching. Eighty-five (85) papers met the study criteria and were published between 2009 and 2019 [62,63,64,65,66,67,68,69,70,71,72,73,74,75,76,77,78,79,80,81,82,83,84,85,86,87,88,89,90,91,92,93,94,95,96,97,98,99,100,101,102,103,104,105,106,107,108,109,110,111,112,113,114,115,116,117,118,119,120,121,122,123,124,125,126,127,128,129,130,131,132,133,134,135,136,137,138,139,140,141,142,143,144,145,146] (Figure 1). Most papers were from sub-Saharan Africa (65): South Africa (17), Kenya (9), Malawi (8), Uganda (7), Ethiopia (6), Ghana (4), Rwanda (3), Tanzania (2), Zambia (2), Botswana (2), Lesotho (1), Liberia (1), Mozambique (1), Niger (1), and one that addressed both Malawi and Ghana. The remainder described mHealth interventions in developing countries overall (6), African countries overall (1), Asian countries (8; including Bangladesh, India, and Pakistan), or Latin America and Caribbean countries (5, including Brazil, Caribbean countries, Guatemala, and Peru).

The study concerned factors that ‘impact’ mHealth adoption, with impact including both positive (enabling) or negative (impeding) effects. A final set of seven primary categories, comprised of seventeen sub-categories, was identified, plus a Miscellaneous grouping (Table 2).

### 3.1. Engagement and Funding

Any project, mHealth or otherwise, requires a variety of stakeholders to participate in the undertaking in terms of their time, effort, skill, or money. Given the spectrum of stakeholders typically involved, this was identified as ‘multi-sectoral engagement’ with two levels of engagement identified at local and higher levels—‘community’ and ‘political’ levels. Funding for mHealth activities is required and can be associated with the political level of decision making, whether in the national or corporate sense. Evidence was seen for each of these impacting adoptions of mHealth initiatives by healthcare workers.

#### 3.1.1. Multi-Sectoral Engagement

This was related to any evidence of how deliberate collaboration between or among the stakeholder groups (e.g., government, civil society, private sector) or sectors (e.g., health, education, finance) impacted mHealth adoption.

The literature showed that organisational and multi-sectoral engagement among institutions is required for effective service delivery and impacts mHealth adoption by health workers [62,63]. Stakeholder collaboration by sharing information and resources creates a supportive setting that facilitates adoption [64,65,66,67,68,69,70]. Similarly, collaboration and coordination of the activities of other service providers by intermediary health service providers (i.e., the district health management system) is particularly valuable and, with appropriate leadership, creates a favourable culture within a well-organised community health structure [71,72,73,74]. Resistance from health workers to teaming up was observed, which would impair adoption [75], with one recommendation being for a shift to a top-down approach when scaling up a project nationally.

Linkage of the community health system with existing community support systems and facility-based services for effective delivery would be conducive [76]. Such linkage amongst allied health systems has the capacity to reduce health worker’s workload, encouraging adoption [77].

In situations where mHealth is to be deployed nationwide, the objectives of the programme must align with the country’s health policy agenda (or eHealth strategy) [66]. Scaling mHealth to a national level would require standardisation of health services and unique patient identification [78] to support follow-ups and effective referrals [77,79], demonstrating that distal factors also impact adoption.

#### 3.1.2. Community Participation and Ownership

Overall, the literature showed that strong community participation and ownership of programmes positively affects healthcare worker adoption of mHealth, as in Cameroon where it was key to the effective adoption of text messages to improve drug adherence [80]. Experience in Botswana showed that for successful mHealth adoption initiatives must be owned, led, and driven by local stakeholders [66], and should be tailored to reflect the local setting [66,68,81].

#### 3.1.3. Political Leadership and Funding

Appropriate leadership and stable funding are inducements to adoption. mHealth programmes in the developing world are often funded by external donors and may only survive the funding cycle, leading some authors to conclude that local government funding and leadership are non-negotiable elements for successful deployment [80,82]. Whilst stable funding is important, a greater impediment was the failure or lack of leadership to encourage healthcare workers to proactively use the information generated or stored for healthcare provision, planning and development [62]. Other barriers included governance constraints such as those arising from resistance to adoption or buy-in from ministries of health [83], poorly informed decision makers regarding the benefits of mHealth [84], and lack of programme alignment with national health and education goals [66]. Management at the various levels of healthcare delivery must show insight, leadership, commitment and ownership towards the promotion of mHealth [62,72,85]. Provision of subsidies to encourage ownership and use of mHealth solutions or enforcement of institutional mHealth protocols was suggested [83], as was mentorships for new district health teams to facilitate effective leadership [86].

### 3.2. Training and Technical Support

The term training refers to four aspects or levels of instruction to ensure effective use of mHealth systems: ‘education’ of a small number of personnel leading to an academic graduate qualification (e.g., MSc, PhD); ‘instruction’ of a slightly larger number of personnel (e.g., to provide proficient network managers); ‘teaching’ of a still larger number of individuals in terms of the use of specific technologies, devices, and services (primarily ‘users’); and ‘awareness’ of the general populace [147]. Technical support addresses ongoing services to provide users with help, advice, and maintenance of mHealth implementations. The literature showed that both training and technical support impact healthcare workers adoption of mHealth.

#### 3.2.1. Training

Providing healthcare workers with relevant training (instruction and teaching) on the use of mHealth systems was considered critical to successful adoption. Aligning with the above descriptions, Petrucka et al. indicated that when introducing mHealth resources to healthcare workers, there is the need to first create awareness before providing the training [68]. The identified literature showed that healthcare workers can effectively deliver mHealth services when they have the requisite information and training [86,87,88,89,90,91,92,93,94,95,96,97], which, as noted below (System Utility, Context and Content), encourages adoption through empowerment and a greater sense of professionalism and accomplishment. When challenges in delivery occurred, they were attributed to a lack of training [70].

Targeted training [83,95,98,99,100,101], e.g., towards those with greater literacy, may be beneficial, as demonstrated in Liberia when both traditional and certified midwives were trained to use an mHealth solution; certified midwives performed significantly better even though both groups received the same training [99]. Similarly, healthcare workers with prior computer literacy were more likely to adopt and use mHealth than those without any computer knowledge at all [95,100]. Such an outcome was not consistent, with no clear differences in responses between Community Healthcare Workers (CHW) and nurses, despite differences in educational background and training [102], and in another case, poor adoption of mHealth occurred despite providing in-depth training to a well-educated user population prior to implementation and with technical support throughout [103]. Papers noted that training was considered positively and could be utilised as an incentive [104], that the demand for training was high [97], that training in local languages was needed [72,73,105,106], and that there was need for organisational readiness to train users [70]. One raised the issue of integration of training into the medical curriculum and accepted practice [84].

The quality of training determined the quality of the service delivered to patients [107], and several papers noted the need for repeated, regular, or refresher training [72,77,108,109,110,111,112], lest gains be lost [109]. Beyond classroom training, mHealth itself was identified as a strategy by which to improve the skills and knowledge of healthcare workers, and establish a trained ‘mHealthcare’ workforce at scale [69,111]. For example, text messaging was a feasible, acceptable and inexpensive approach to improving healthcare worker performance, and served as reminders of classroom training content and good field practices to those who had attended classroom training, but also helped spread information to those who had not [69]. CHW also expressed a desire for continuing medical education using distance learning [113].

Training time must not be too long [105,114], yet not be too short, insufficient, or inadequate [92,93,115], and should have dedicated time that does not interfere with daily tasks [100]. The efficiency of mHealth use by healthcare workers was noted to depend on factors such as level of competence, availability of the appropriate technology, and the frequency of use of the device [116,117].

#### 3.2.2. Technical Support

This refers to availability of developmental support (e.g., clarity on a deployment issue or needs), plus helplines, physical repairs, and replacement of damaged devices for deployed mHealth implementations. Each of these were identified as factors impacting healthcare worker adoption of mHealth solutions [72,74,78,83,91,92,107,110,118]. Ideally there should be support units dedicated to the maintenance and replacement of mHealth devices [72,119].

### 3.3. Infrastructure

This refers to the basic physical and organisational structures and facilities required for the optimal deployment and operation of an mHealth enterprise. It stretches from availability of mobile devices to healthcare workers, through wired and wireless networks for connectivity, to national grids for providing electricity to power each component. Without these, mHealth would be impossible. Even issues as simple as mobile phone availability can become a barrier. In one country, the introduction of the latest and newer version of a mobile phone led to the desired mobile devices being unavailable [63].

#### Availability and Reliability

Successful and sustained mHealth requires a reliable and effective network technology [70,95] that must be maintained and upgraded to deal with emerging challenges [62,77,78,103]. Technological barriers were identified as a major impediment to mHealth adoption [63,83,86,88,90,120,121]. Certainly, infrastructure limitations exist for electricity, mobile service coverage, and Internet access, which may only be available intermittently or not at all in certain areas [94] and directly impact mHealth implementations [101]. Improving mobile network coverage, particularly in resource poor settings, is essential [65,79,122].

Even the perception of a reliable and regularly updated mHealth system promotes adoption [109]. Of concern, therefore, is that infrastructure failures are often beyond the control of mHealth implementers, yet they reflect immediately and negatively on the effectiveness and reliability of the mHealth system [112,120,123]. This has led to the recommendation for both online and offline capabilities for mHealth solutions [77]. Issues with a lack of reliable network coverage or power preventing the use or charging of mobile devices were the most common infrastructure issues noted [62,63,66,71,77,81,83,86,91,93,95,97,99,101,102,104,108,110,111,112,113,115,118,122,123,124,125,126], although this was not universal with one study reporting wider service coverage than expected [79] and another with coverage reaching even the most remote areas [72].

### 3.4. System Utility

This term was used to refer to issues related to overall usefulness and helpfulness and was viewed in terms of its context and content and its flexibility as a communications mode, but also accommodated issues related to privacy, confidentiality, and security as well as socio-cultural issues.

#### 3.4.1. Context and Content

Context was used to refer to how circumstances reflecting the working environment impacted mHealth adoption by healthcare workers. Content referred to the information content of the system and its alignment with and value to the normal working environment of a healthcare worker that would affect adoption.

Papers frequently noted the importance of healthcare workers believing the mHealth interventions reinforced or improved their own impact on patients. This was through improving their services and providing better care [83,107,108,116], improving the efficiency with which they provided the services [64,65,79,103,119], increasing their referral rates [127], maintaining their skills [116], overcoming geographic or weather-related barriers [89,97], accessing and allowing dissemination of up to date information [95,118], offering administrative advantages over traditional ‘paper-based’ approaches [102,128], raising their image, social standing, confidence, and morale [73,115], or showing cost benefit [65,129]. Some, however, questioned if it threatened their job security [128,130], or were apprehensive about increased work load, reduced remuneration, and intrusive supervision [110]. In terms of the design of the intervention, utility in terms of ease of use was noted [81,102,131], with Crumley and colleagues stating “speed, reliability and user-friendly design are non-negotiable for success” [132]. Several papers also noted the importance of involving users in the design process and responding to their feedback [74,77,82,91,107,112,118,133,134,135] and ensuring alignment with user needs and clinical tasks [74,126,130,134,136] which would contribute to user friendly design. One noted the difficulty in gaining feedback since the intervention was abstract to the users [110].

The issues affecting healthcare workers’ acceptance of mHealth cut across technological, social and contextual issues and included culture [27], design architecture [112], flexibility [107,137], user satisfaction [114], content management [77], and features enhancing value and supervision of the healthcare workers [110]. Healthcare workers in Botswana were satisfied using SMS technology to query international health databases and access national treatment guidelines because they considered it friendly and easy to use [131]. Other authors advised that mHealth be developed within the framework of designs that healthcare workers are already comfortable and familiar with [88,124], often termed ‘user-centred’ [74,81,132,133,134], reflecting local context and enabling local capacity building to handle future challenges and scalability [77].

#### 3.4.2. Flexible Communication Mode

Improved communication has the potential to translate into better health outcomes for patients [67]. The ability to easily communicate with patients, and/or with colleagues on behalf of patients was highlighted, together with the value of inter-collegial communication to facilitate the process of providing healthcare [62,135]. For example, establishing a mobile phone network for communication between village health volunteers and supervisors increased monthly report submissions and more timely consultations, encouraging adoption. Experience from South Africa, Kenya, and Cameroon showed two-way communication was the desired choice for patient-health worker communication [80,106,121]. It was suggested that interactive and real-time / near real-time communication (whether voice-to-voice, video chatting, text chatting platforms, or instant messaging) would promote further adoption and use [93].

It was noted that enhanced communication must not unnecessarily increase the workload of health workers [107], and can be valuable in rural and remote regions where significant distance separates facilities, healthcare workers, and patients [75]. Modes of communication included voice, SMS, or Internet based communication. An increasingly popular instant messaging platform, WhatsApp, presents a much cheaper means of interactive communication but requires smartphone technology restricting its use in resource poor settings [83]. As a viable alternative, SMS text messages were an effective and available interactive mode of communication used by healthcare workers within mHealth systems [27,109,138] and can also facilitate data gathering [139].

#### 3.4.3. Privacy, Confidentiality, and Security

Within healthcare, privacy relates to the ability of a person to decide for themselves what, when, and how any personal health information (PHI) is shared with or communicated to other healthcare workers. Confidentiality relates to ensuring PHI remains private, with access only to those authorised to access it by the patient. In turn, security relates to the steps taken to prevent unauthorised access to PHI within computers, mobile phones, databases, or websites. The need for privacy, confidentiality and security now includes and is amplified by electronic transmission of personal health information through mHealth [126], and concern has been expressed [69]. This can be a challenge to the broad adoption of mHealth due to differing norms for privacy, confidentiality, and security within communities and ethnic groups [94,115,119,136,140]. For example, health workers in Uganda using mHealth to manage stroke initially filmed their patients but found this caused discomfort and ‘changed the mood’ leading to filming being abandoned [74]. Another study showed the need for anonymity in an SMS messaging group setting due to stigma related to HIV status [139].

#### 3.4.4. Socio-Cultural Issues

Social context refers to the issues of gender, age, ownership of mobile phones, language, educational level, etc., and their implications to mHealth adoption by healthcare workers. It was noted that cultural misalignment (initial negative perceptions of the technology by new users prior to any training) could be detrimental [66]. Socio-cultural issues are of crucial concern in the introduction of mHealth in the developing world [125,137]. For example, in Kenya introducing cultural and gender sensitive SMS messages markedly improved the involvement of males in prevention of mother-to-child transmission of HIV/AIDS [76]. Social factors such as age, language barriers, educational level, care-giving experience, preference in data capturing tool, distance from healthcare workers to patients, and mobile phone ownership are important issues to consider in rolling out mHealth services among healthcare workers [118]. Differences in culture between developed and developing countries were identified in three areas: ‘individualism versus collectivism’, ‘power distance’, and ‘masculinity versus femininity’, with developing countries leaning towards collectivism, comfort with power differentials, and preferential treatment towards males [27].

### 3.5. Healthcare Worker’s Mobile Phone—Cost and Ownership

This refers to how the cost, ownership and maintenance of mobile phones and their accessories affect the adoption of mHealth among healthcare workers.

#### 3.5.1. Mobile Phone Ownership

Possession of a mobile device by healthcare workers is a prerequisite for mHealth deployment among healthcare workers (HCW) [91,140]. In some studies, smartphone ownership was universal [84,87,113,114,138] or almost so [90,95,130], although in others no participant owned a mobile phone [110], or mobile phones were provided (with participants allowed to use them for personal use and communicating with other HCW and district officials which motivated the healthcare providers) [62]. In one study, the type of mobile phone was noted, with 57% being feature phones, 34% being basic phones, and only 9% being “smartphones” [138].

#### 3.5.2. Affordability and Cost

It was reported that healthcare workers would adopt mHealth if they consider it to be cost saving [129,141] and affordable [138], although affordability differs with nurses being less able to adopt the use of mobile phones being poorly paid compared to doctors [100]. The chosen devices should not be complicated or expensive but durable, simple, and as ‘low-end’ as feasible for the mHealth solution [77]. The biggest barrier to adoption and use in some instances was the cost of data [87], and in one setting, ongoing use of an mHealth solution was stated to be dependent on the care givers’ and patients’ willingness and ability to absorb ongoing costs, in contrast to an expected government or sponsor-supported service [142]. In some settings, airtime costs for voice and text communication on mobile phones were considered barriers [83,84], or at least to limit their use [102]. In others cases, healthcare providers informally used their personal phones in the work place and bought airtime and data at their own expense [71,100], which was suggested to increasingly be an expectation [69] prompting suggestions for reimbursement models or subsidies [87,123]. It was further noted that willingness to use mHealth does not imply preparedness to incur any adoption or sustainability costs [142].

#### 3.5.3. Phone Charging Issues

The ability, or inability, to easily charge mobile phones impacts mHealth adoption by healthcare workers. Access to electricity to charge phones was recognised as a challenge [83,104,119], with healthcare workers sometimes having to travel long distances to do so [120] or using car batteries [67]. For mHealth to be broadly adopted, battery charging must be resolved in innovative ways [93,120,121,125].

### 3.6. Motivation and Staffing

Issues such as availability of personnel, appropriate workload, and motivation were identified as factors affecting the adoption of mHealth by healthcare workers.

#### 3.6.1. Adequate Personnel and Workload

Availability of an adequate number of well-trained healthcare workers, together with an appropriate and reasonable workload, affects healthcare workers’ adoption of mHealth [75]. Adequate personnel has been addressed under ‘Training’. Research in Brazil showed an increased workload due to mHealth since it introduced new additional responsibilities (transfer to a mobile app of previously recorded data) [143]. Some healthcare workers resented the lack of compensation for the increased workload [97], which was considered a burden for rurally based nurses [112]. In some settings, the additional workload was considered unnecessary and inappropriate [107], and concern was raised about the possibility of a burgeoning number of apps within any specialty (versus a single streamlined application) leading to increased workloads [132]. There are also situations where disparate and uncoordinated mHealth apps may inadvertently overlap and increase workload [87]. Situations certainly arise where healthcare workers may have to be contacted by patients during evenings and weekends, invading their private space and time [71]. There is, therefore, a need to avoid overburdening healthcare workers to encourage adoption [100,128].

#### 3.6.2. Healthcare Worker Motivation and Commitment

A healthcare worker’s motivation (intention to start a task) and commitment (intention to complete a task) impact adoption of mHealth interventions and can be influenced through incentives. To some degree, mHealth is self-motivating because it is seen as the future of medical practice and learning to use it is empowering and raises credibility, self-confidence, and social standing [84,102,103,108,115,119]. One paper suggested it transforms healthcare workers from passive recipients to active information seekers and providers [79]. In contrast, specific incentives (e.g., financial) may be required [74,126,144], particularly for volunteer workers [92]. In Pakistan, physicians may have shown variable use of mHealth because of the absence of a financial incentive [145]. Occupational therapists in Uganda using mHealth to support stroke management were reluctant to use the system until financial incentive contracts were signed with them before the commencement of the programme [74]. When motivation is purposefully and consciously provided, some believed it would ‘win the hearts and minds of the health worker’ [103].

Availability of training can be considered an incentive also, and insufficient training lowers healthcare worker motivation [92]. Motivation can take several forms: replacing faulty or lost mobile devices [71,102,110], or providing monetary rewards [74], free airtime [71,78,83], training [92,102], and knowledge on case management [144]. Providing appropriate motivation for healthcare workers is challenging [92], but will enhance adoption [72]. Whilst incentives may be necessary, they alone may not be enough to overcome negative socio-cultural practices [75]. In addition, the issues of job security and over-expectation of mHealth’s abilities may impede adoption [129,130].

### 3.7. Patient’s Mobile Phone—Cost and Ownership

Whilst mHealth may require healthcare workers to possess a mobile phone, the need for patients to do so may be less obvious. Increasingly healthcare workers are communicating directly with patients electronically, necessitating both parties to possess mobile devices, typically mobile phones [100].

#### 3.7.1. Mobile Phone Ownership

Mobile phone ownership among patients was found to be a determinant of mHealth adoption among health workers [83,119], with ownership being described as variable but with most “not own(ing) phones themselves” although many “leveraged” access (borrowing; local kiosks) [119]. A 2019 report from the African Region stated “patients not having access to phones (whether through lack of money or not owning their own phone)” was a barrier [83]. Furthermore, where there were community phones available to communicate with health facilities, cost, poor road, and poor telecommunication networks were major challenges affecting their use [119,125]. In addition, lack of ownership led to sharing of phones with others, which can breach patient privacy and confidentiality [71,88,120].

#### 3.7.2. Affordability

The most obvious barrier to mobile phone ownership was cost [83]. This was supported in a rural Bangladesh study of maternal and neonatal mothers who, with low or no income, could not access mHealth resources [64]

### 3.8. Miscellaneous Factors

Of note were other non-categorised factors.

Whilst mHealth tools can strengthen a programme [72], they cannot salvage it and so should not be used in an attempt to fix an already broken system [104].Nontechnical challenges are more difficult to resolve than technical challenges [122].Technical support and repair services could be developed through leveraging of local expertise and open-source resources for application development [77].‘Reminder fatigue’ was identified as an inhibiting factor, with clinicians no longer reacting to automatically generated reminders due to over-use [144,145].The need for selection of simple, appropriate technology solutions was noted [141].mHealth solutions must be compatible with the complete range of mobile phone devices (low-end basic phones to high-end smart phones) [124,146], a spectrum of which will always exist due to affordability and constant technological advances.Although mobile phone adoption and use is influenced by many factors, their adoption also influences technological, personal, organisational, and cultural factors [125].Staff turnover can compromise adoption through loss of key development and support staff [77].Concern for theft of mobile devices, even feeling vulnerable carrying them, was noted [90,104,118,130].

### 3.9. Proposed Framework for Healthcare Workers

Resources identified through the search typically addressed ‘mHealth adoption’ issues that could be considered to broadly relate to healthcare workers; however, specific ‘healthcare worker mHealth adoption’ issues were not addressed. Consequently, healthcare worker related issues were often teased out from review of individual papers and then categorised. To facilitate application of the insight gained around mHealth adoption factors for healthcare workers in developing countries, a conceptual framework was constructed (Figure 2). For healthcare workers in developing countries to successfully adopt mHealth the issues identified in the framework must be considered within the context of any specific setting and assessed and addressed as necessary. Whilst factors in each category have been shown to individually and directly impact healthcare worker mHealth adoption (bi-directional arrows), possible indirect and collective impacts cannot be discounted (linking of categories). For example, as ‘infrastructure’ is enhanced and higher bandwidth wireless capability is extended to rural and remote areas, it is likely greater exposure to mHealth applications will occur, leading to greater willingness to ‘engage’ and ‘motivation’ to adopt mHealth solutions. Similarly, it can be visualised that as technology advances ‘cost’ of mobiles will decrease, and ‘ownership’ of more capable smart phones will increase, leading to the implementation of more practical mHealth solutions with greater ‘system utility’ The potential combinations and permutations are great.

## 4. Discussion

This study has identified seven specific categories and 17 sub-categories of factors that impact the adoption of mHealth by healthcare workers in developing countries; the proposed, evidence-based conceptual framework incorporates all of these. Unlike theoretically based studies (e.g., using TAM and UTAUT models), this study did not presume any factors beforehand and instead used empirical research of actual mHealth applications to identify relevant factors. Overall, evidence indicates that mobile technology tools, such as smartphones, feature phones, and basic phones, substantially benefit healthcare workers, their patients, and healthcare delivery, encouraging adoption [1,2,3,4,5,6,7,8,9,10,11,12,13,14,15,16,17]. The seven themes impacting adoption and application of mHealth by healthcare workers in the developing world are: Engagement and funding, training and technical support, infrastructure, system utility, motivation and staffing, and the cost and ownership of a mobile phone for both patients and healthcare workers.

Though relatively new to healthcare, mHealth is already changing the way healthcare workers collect, access, disseminate, and manage health information either at the patient or facility level. The major challenge in the developing world has been identifying the facilitating and impeding factors that affect its successful adoption among healthcare workers. mHealth, although not the solution to all the problems affecting healthcare workers, provides significant opportunities to improve access, reduce cost of accessing care, promote patient-provider interaction, improve disease management, and support education and training. Understanding the full spectrum of issues that impact mHealth adoption, and enhancing those that enable whilst minimising those that impede adoption, is paramount to successful adoption.

Using empirical evidence, this study has highlighted the different factors that have impacted the adoption of mHealth interventions by healthcare workers in the developing world. The findings align well with other studies, although not focussed on mHealth [148,149]. The proposed conceptual framework was developed from the seven categories of factors identified. This diagrammatic view of the adoption factors, together with their specific sub-factors, can be used to inform future efforts to encourage healthcare workers in the developing world to adopt mHealth solutions. Successful adoption and ongoing use require these factors to be considered and addressed in each setting before implementing any mHealth project for healthcare workers in the developing world.

A broad spectrum of factors has been identified. The need for multi-sectorial engagement among departments and agencies responsible for the facilitation and coordination of the service and sustainable funding is considered critical for healthcare workers mHealth adoption. This has implications for collaboration among allied institutions whose duty it is to provide certain services within support architecture [62], strong community participation [66], as well as political leadership and funding [74]. Multi-sectorial engagement and sustainable funding may well be important for mHealth success, but the lack of appropriate leadership to ensure that health workers use the system to achieve the necessary objective is considered the biggest challenge to adoption [62].

The acquisition of the appropriate training and the availability of the relevant technical support was identified as among the core items required by healthcare workers to make them successful in the use of mHealth. This has implications for improving curricular content for trainee healthcare professionals and also providing adequate training for current healthcare professionals. This may require policy decisions in curricular development at health educational institutions to develop competencies in the effective use of mHealth [84]. There is also the need to identify mHealth competency gaps of current healthcare professionals and develop programmes to address them. In addition, technical support (e.g., such as helplines, mobile phone repairs, replacement of damaged devices) need to be provided [83,92].

System utility issues can be limiting factors. In order to improve adoption, healthcare workers should be involved in the design process of mHealth interventions, including content and context. Furthermore, an efficient and reliable communication system between the health workers is required, and privacy and socio-cultural issues need to be addressed. Infrastructural barriers (i.e., poor or inefficient networks system and power supply) remain major impediments to mHealth adoption by healthcare workers [63,83,121].

The costs of mHealth, including the cost of ownership, maintenance, and use of the device, must be considered. Device ownership is a prerequisite for mHealth implementation [91,140], and consideration may have to be given to device provision or subsidisation. In situations where there is the need to pay for certain services in using mHealth, healthcare workers have been willing to do that provided it is cost saving [129,141], and affordable [138]. Similarly, mobile phone ownership by patients affects healthcare workers’ adoption of mHealth, especially when offering teleconsultations and providing message-based services.

Whilst the reviewed papers suggested poor ownership rates [119], more recent publications suggest varied ownership rates, gender differences, and differences in type of phone owned (basic vs. feature vs. smartphone). For example, overall ownership varied between ~50% to almost 100% across 15 developing countries, with gender gaps being most marked in Pakistan (56%), Ethiopia (25%) and Nepal (24%), and generally being worse for rural and poor women [150]. That study concluded, “inequalities persist by gender, geographical areas and sociodemographic characteristics”. Two studies of diabetic patients in Nigeria a decade apart showed currently 97% owned a mobile phone (39% basic phones, 61% smartphone), although ‘few’ had access to Internet services [151], compared to ~68% ownership in another Nigerian study a decade earlier [152]. However, cautionary comments were also made, suggesting estimates can be inaccurate (e.g., users with more than one active SIM card being counted as ‘unique subscribers’ [153], and ‘access’ through sharing of mobile phones being fraught with confidentiality and privacy issues [154].

Staffing and motivation was also seen to impact the successful adoption of mHealth by healthcare workers. There is need to implement an effective change management process, ensure appropriate training and obtain staff buy-in to improve mHealth adoption, supplemented with identification of “motivational triggers”[92]. These align with earlier recommendations from the European Union [155]. mHealth may undoubtedly add an extra layer of workload to the health worker’s duties [87] and incentives should be considered as extrinsic motivation [72]. These have included replacing faulty or lost mobile devices [71,102,110], giving cash rewards [74], providing free airtime [71,78,83], training [102], and assisting in case management [144].

There is strategic need to maximise use of mHealth initiatives in developing countries. The Sustainable Development Goals (SDGs), intended to end poverty, combat climate change and fight injustice and inequality by 2030, include Goal 3—good health and well-being for all [156]. Given the constrained healthcare workforce and growth in mobile connectivity throughout the developing world, mHealth is anticipated to become a critical component in achieving SDG-3 and perhaps other SDGs. It has been reported there are already over 1000 mobile health services in low-income countries providing health content and diagnostics services [74], and the World Health Organisation actively promotes the use of mHealth [54,157].

Whilst the COVID-19 pandemic has increased the application of mHealth interventions, this has, of necessity, occurred rapidly (with little or no consultation or pre-testing) [158] and only after favourable re-casting or exempting of prior restrictive legislation or guidelines (of uncertain duration) [159], despite concern for bioethical and legal issues [160], and with few evaluations reported. Furthermore, many pre-pandemic issues remain unresolved, including technical infrastructure, smartphone access, language issues, patient satisfaction and safety issues, privacy issues, increased inequalities in health provision, and desirable shifts from former practice [161]. Although post-pandemic use of mHealth is anticipated, new or ongoing issues and concerns have been noted [162]. It is likely that negative results or unsustainable implementations stemming from the rapid implementation period will occur, necessitating reflection and application of an enhanced understanding of factors known to impact mHealth adoption. The findings of this study will inform that process.

### Limitations

The use of only two databases, PubMed and Scopus, and the selection of only peer reviewed papers in English may have limited the scope of resources for analysis. The proposed framework may thus not address all possible factors identified in the subcategories of the seven themes that influence health workers’ adoption of mHealth.

## 5. Conclusions

Application of the framework and accompanying subcategories will raise awareness of practical issues that must be addressed to ensure successful adoption of mHealth solutions for healthcare workers in the developing world. This will help shape future policy and practical approaches to mHealth implementation and use in the developing world for healthcare workers, thereby increasing adoption.

## Figures and Tables

**Figure 1 ijerph-20-01244-f001:**
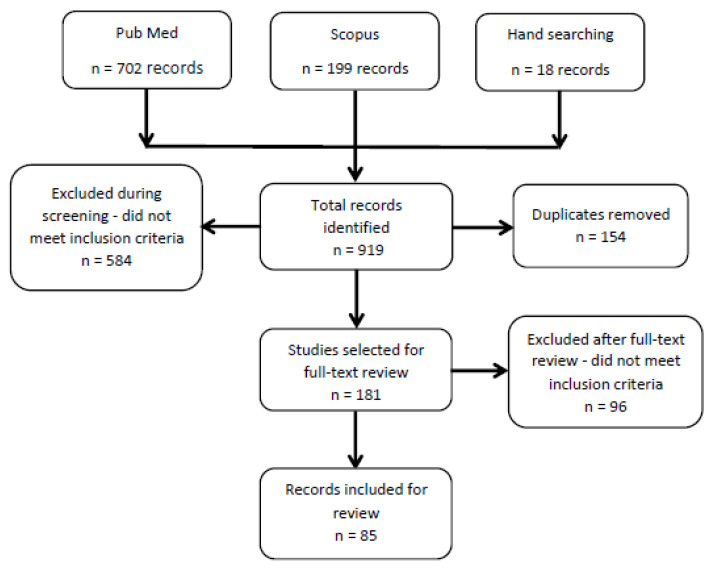
PRISMA flowchart for literature searches.

**Figure 2 ijerph-20-01244-f002:**
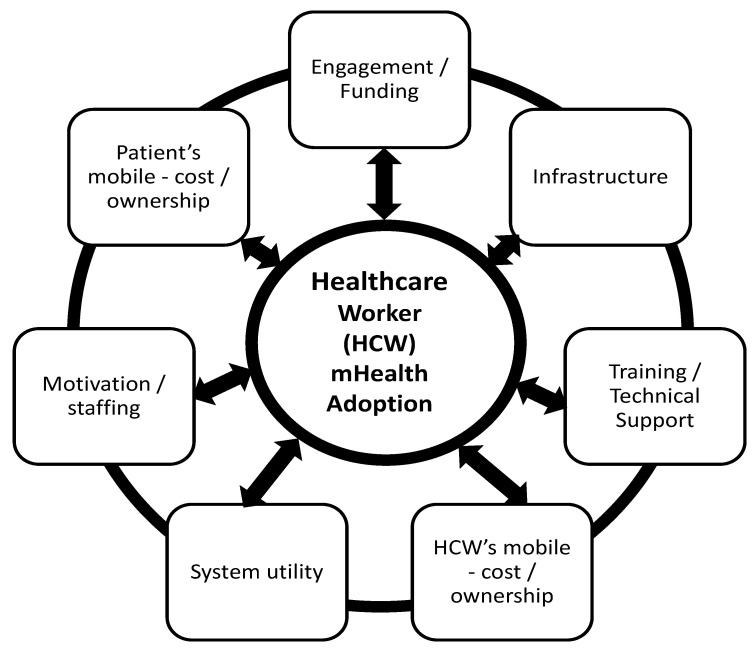
Proposed conceptual framework for successful adoption of mHealth by healthcare workers in the developing world.

**Table 2 ijerph-20-01244-t002:** Findings from 85 included publications. Check-mark indicates content relevant to one of seven Primary Categories or 17 sub-categories (miscellaneous sub-category discussed in the Results). HCW = Healthcare Worker.

Primary Categories	Engagement and Funding			Training and Technical Support		Infrastructure	System Utility				HCW—Mobile Phone Cost and Ownership			Motivation and Staffing		Patient—Mobile Phone Cost and Ownership		
Sub-Category	Multi-Sectoral Engagement	Strong Community Participation and Ownership	Political Leadership and Funding	Technical Service and Support	Training and Education	Availability and Reliability	Context and Content	Flexible Communication	Privacy, Confidentiality, and Security	Socio-Cultural Issues	Mobile Phone Ownership	Affordability and Cost	Phone Charging Issues	Adequate Personnel and Workload	Motivation and Commitment	Mobile Phone Ownership	Affordability	Miscellaneous
Author & Year																		
Madon et al., 2014 [62]			√			√					√							
Pérez et al., 2014 [63]						√					√							
Huq et al., 2014 [64]							√										√	
Lemay et al., 2012 [65]						√	√											
Littman-Quinn et al., 2013 [66]		√	√			√				√								
Nonaka et al., 2014 [67]								√			√							
Petrucka et al., 2013 [68]		√			√													
Rassi et al., 2018 [69]					√				√			√						
Stroux et al., 2014 [70]					√	√												
Mehta et al., 2018 [71]						√						√		√	√	√		
Ngabo et al., 2012 [72]			√	√	√										√			√
Ramukumba et al., 2019 [73]					√		√											
Teriö et al., 2019 [74]				√			√		√						√			
Ayiasi et al., 2015 [75]								√						√	√			
Jennings et al., 2013 [76]										√								
Shiferaw et al., 2018 [77]					√	√	√											√
Medhanyie et al., 2015 [78]				√		√									√			
Campbell et al., 2014 [79]						√	√								√			
Georgette et al., 2016 [80]		√	√					√										
Källander et al., 2013 [81].		√				√	√											
Lindberg et al., 2019 [82]			√				√											
Allsop et al., 2019 [83]			√	√	√	√	√	√				√	√		√	√		
Barnor-Ahiaku, 2016 [84]			√		√						√	√			√			
Kaonga et al., 2013 [85]			√															
Karari et al., 2011 [86]			√		√	√												
Fischer & Sebidi, 2019 [87]					√						√	√		√				
Ginsburg et al., 2015 [88]					√	√	√									√		
Hirsch-Moverman et al., 2017 [89]					√		√											
Khalala et al., 2013 [90]	√				√	√				√								
Little et al., 2013 [91]	√			√	√	√	√				√							
Musabyimana & Ruton, 2018 [92]	√			√	√										√			
Mwendwa, 2016 [93]	√				√	√		√					√					
Shovlin et al., 2013 [94]	√				√	√			√									
Stanton et al., 2015 [95]	√				√	√	√				√							
Tumusiime et al., 2014 [96]	√				√													
White et al., 2016 [97]	√				√	√	√							√				
Lamanna & Byrne, 2019 [98]	√				√													
Perosky et al., 2015 [99]	√				√	√												
Watkins et al., 2018 [100]	√				√							√				√		
Woods et al., 2012 [101]	√				√	√												
Bhatt et al., 2018 [102]	√				√	√	√					√			√			
Chaiyachati et al., 2013 [103]	√				√	√	√								√			
DeRenzi et al., 2011 [104]	√				√	√							√					√
Medhanyie et al., 2017 [105]	√				√													
Zakus et al., 2019 [106]	√				√			√										
Chirambo et al., 2018 [107]	√			√	√		√	√						√				
Agarwal et al., 2015 [108]					√	√	√								√			
Jones et al., 2012 [109]					√	√		√										
Modi et al., 2015 [110]				√	√	√	√				√				√			
Shah et al., 2018 [111]					√	√												
Vedanthan et al., 2015 [112]					√	√	√							√				
Nilseng et al., 2014 [113]					√	√					√							
Surka et al., 2014 [114]					√		√				√							
Coetzee et al., 2018 [115]					√	√	√		√						√			
Thomsen et al., 2019 [116]					√		√											
Gouse et al., 2018 [117]					√													
Shozi et al., 2012 [118]				√		√	√			√								
Chang et al., 2011 [119]				√			√		√		√		√		√			
Hamainza et al., 2014 [120]						√					√		√					
Laktabai et al., 2018 [121]						√		√					√					
Medhanyie et al., 2015 [122]						√												√
Mannik et al., 2018 [123]						√						√						
Alberts et al., 2014 [124]						√	√											√
Manda & Herstad, 2010 [125]						√				√	√		√					√
van Heerden et al., 2017 [126]						√	√		√						√			
Martinez et al., 2018 [127]							√											
Brandt & Hidalgo, 2019 [128]							√							√				
Chang et al., 2013 [129]							√					√						
Chang et al., 2013 [130]							√				√				√			
Armstrong et al., 2012 [131]							√											
Crumley et al., 2018 [132]							√							√				
O’Connor et al., 2017 [133]							√											
Mburu & Oboko, 2018 [134]							√											
Vélez et al., 2014 [135]							√	√										
Tariq & Akter, 2011 [136]							√		√									
Mall et al., 2013 [137]							√			√								
Zurovac et al., 2013 [138]								√			√	√						
Dean et al., 2012 [139]								√	√									
Nhavoto et al., 2017 [140]									√		√							
Mahmud et al., 2010 [141]												√						√
Hwabamungu & Williams, 2010 [142]												√						
Rajan et al., 2016 [143]														√				
Kaunda-Khangamwa & Steinhardt, 2018 [144]															√			√
Hashmi & Khan, 2018 [145]															√			√
Manjunath et al., 2011 [146]																		√

## Data Availability

The data that supports the findings of this study are available on request from the corresponding author without restrictions.

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
