# Peer review of "Healthcare Workers’ Perspectives of mHealth Adoption Factors in the Developing World: Scoping Review"

_ijerph, 2023, doi:10.3390/ijerph20021244_

Round 1

Reviewer 1 Report

Major comments:

This is an important topic and of high interest and relevance. The manuscript is very well written overall, although the overall structure could be enhanced to better guide the reader. There seem to be two main aims of the paper: to conduct a scoping review identify to what extend mHealth platforms have already been used in the developing context, and to use the results of the scoping review to create a framework to guide mHealth implementation in developing contexts. I would improve the abstract and background to clearly highlight these two separate aims. One big criticism, however, is that the scoping review was conducted in January of 2020, which is now almost three years ago. This is a long time in the context of the fast-moving world of mHealth, and I strongly recommend updating the scoping review to include more recent studies.

The aims and reporting of the review are somewhat inconsistent. I suggest that the authors specifically reference and report a set of guidelines such as the PRISMA guidelines for scoping reviews to enhance the quality of their paper: https://www.acpjournals.org/doi/10.7326/M18-0850.

The methodology for the development of the framework for implementation needs to be described further. There is enough information about how the scoping review was conducted, but then the results were almost entirely the framework analysis.

Specific comments:

Abstract

-First line is not quite native flow English. Would say instead something like, “mHealth applications provide health practitioners with platforms that enable disease management, facilitate drug adherence, …”

-In second line, I would avoid the use of consequently. I doubt that countries are investing in telecommunications only because of its use in healthcare, and the sentence stands fine on its own.

-The research question for the scoping review is not clear as written in the background. It sounds like instead of a scoping review a new methodology (“framework”) is being proposed? It would greatly help the reader to see what the specific driving question of the scoping review is here.

-Results and discussion: it would help to at least briefly summarize some of the factors identified. The methods could be shortened to do so, as the methods of a scoping review are well established.

Introduction

Lines 53-57: suggest expanding this section and certainly having more references, as mHealth in the developing context is the entire point of this paper. What are the barriers to mHealth implementation specifically in the developing context? What are the advantages to mHealth specifically in this context? I would move this discussion down into the research gap paragraph in lines 88-98, as this is the central idea of your paper.

Lines 88-98: As stated above, I would incorporate more discussion of mHealth specifically in the developing context here and why it is important.

Table 1: I do not understand what this table is intended to convey. For example, is it reported %, % variance, and what is reference? Are these separate columns? Please add more text or even rethink including it unless you want to specify the factors that affect behavioral intentions, which would be more useful to the reader.

Lines 99-100: I would be careful here. You are conducting a scoping review, not necessarily examining empirical evidence. I would specify that instead.

Line 201: framework…for implementation of future mHealth studies, correct? I would specify that.

Methods

Line 106: Performed in January 2020…that is almost three years ago now. In the context of mHealth that is a long time. Would also state which guidelines you followed for the scoping review (PRISMA…?)

Lines 110-120: I know it makes it very long, but I would consider expanding the search terms to include individual developing countries.

Lines 131-134: I would have this as its own paragraph as this is the discussion of a separate methods aim than the scoping review. I would also expand this discussion. I know you have a reference, but there is only one sentence on how you designed the framework for mHealth implementation and it is the main part of the results. What exactly was the methodology used? How did you decide on the number of categories and subcategories?

Results

Table 2 and all results: this is by far the most interesting contribution of the paper, but I am still not sure how the authors arrived at these identified themes. I would also suggest having the headings be horizontal and not vertical as it is difficult to read these very important conclusions (you could always switch the headings and rows as the headings are the more important result of this table).

Lines 210-220: what is the difference between the training theme identified here at 3.5 and at 3.6? Surely this should fall under one heading?

Discussion:

Lines 529-531: where is the evidence on this benefit presented in the paper? There should definitely at least be a citation here.

-overall discussion: this is largely a restatement of what is written in the results section. Any concrete implications for future mHealth implementation?

-overall throughout paper: editor should have a look at comma usage for readability

Reviewer 2 Report

This structured literature review on HCWs perspectives on mHealth adoption factors in the developing world is clear, well-written and contributes to the field. The references are relevant and thorough and the analysis of the literature is sound. The seven primary categories and 17 sub-categories in the proposed framework accurately capture the findings from the literature. Overall, this article is a positive contribution to the field. 

Minor comments:

The introduction discusses TAM and UTAUT, however, there is no mention of the Diffusion of Innovation which is widely referenced in relation to the adoption of technology in the health domain. It would strengthen the article to include this. 

Table 2 - very thorough. Minor suggestions - add a column separator to make it easier to see what sub-categories are ticked. Also, as the table extends to multiple pages, I suggest locking the title row so it appears on each page. 

Query re sub-category on Trainig and Training and Tech Support. Should it be Section 3.6 or 3.5.1? Similarly, should 3.7 be 3.5.2?

Line 234, regarding computer literacy in the developing world. Does the diffusion of technology in the developing world impact the number of 'digital natives', and is this potentially more relevant than education background?

Formatting issues in Section 3.8. 

Round 2

Reviewer 1 Report

The aims have been more fully clarified, which helps the reader a great deal. Many of my original points were not addressed, however. For example, the authors state that they used, "inductive iterative content analysis" to arrive at their framework, which should be added as well to the abstract. However, this is a very general description of qualitative methodology. It remains almost impossible to follow the process the authors followed to arrive at their framework. What was the coding process like? How many iterations were conducted? How were the categories refined over each iteration? Did only one person do the coding, or two, or three? How were discrepancies in coding addressed? 

lines 82-83 introduction: the addition of this single line about mHealth in the developing world detracts from the flow of the introduction and should certainly be cited. Suggest either deleting or expanding into a proper paragraph. Why is there no reference to the extensive literature specifically around mHealth in developing countries (e.g. DOI: 10.1146/annurev-publhealth-052620-093850)? That is a major gap in the introduction as it stands.

The reasons for not extending the scoping review are noted but should then be written somewhere in the paper. 

Author Response

Response to Reviewer Document
